# Chemometric Evaluation of Official and Advanced Methods for Detecting Olive Oil Authenticity in Canned Tuna

**DOI:** 10.3390/foods14152667

**Published:** 2025-07-29

**Authors:** Marjeta Mencin, Milena Bučar-Miklavčič, Maja Podgornik, Nives Ogrinc

**Affiliations:** 1Department Environmental Sciences, Jožef Stefan Institute, Jamova 39, 1000 Ljubljana, Slovenia; marjeta.mencin@ijs.si; 2Science and Research Centre Koper, Garibaldijeva 1, 6000 Koper, Slovenia; milena.bucarmiklavcic@zrs-kp.si (M.B.-M.); maja.podgornik@zrs-kp.si (M.P.)

**Keywords:** olive oil, canned tuna, authenticity, fatty acids, sterols, CSIA, IRMS, PLS-DA

## Abstract

This study evaluated the authenticity of olive oil in canned tuna products from the Slovenian market using both official methods, including fatty acid (FA) profiling, determination of the equivalent carbon number difference (ΔECN42), and sterol analysis, and an advanced method: stable carbon isotope analysis (δ^13^C) of FAs obtained through compound-specific isotope analysis (CSIA). Results from both methods confirmed that all 10 samples were authentic, as per the limits set by EU Regulation 2022/2104 and supported by the scientific literature. Method performance was further evaluated by adulterating the olive oil from the canned tuna with 5–20% vegetable oil (VO) or hazelnut oil (HO). While FA analysis struggled to differentiate adulterants with similar FA profiles, CSIA of FAs significantly improved detection. However, distinguishing between VO and HO blended samples remained challenging. PLS-DA analysis further supported the potential of using δ^13^C values of FA for food authentication. Storage of adulterated samples also influenced FA composition, leading to significant changes in MUFA/PUFA ratios and δ^13^C values, which became less negative, likely due to oxidative degradation. In summary, the combination of official and advanced methods, supported by chemometric analysis, offers a robust approach to ensuring the authenticity of olive oil in canned tuna.

## 1. Introduction

In recent years, the consumption of fish and fish products has increased due to growing awareness of their health benefits, primarily attributed to their high content of polyunsaturated fatty acids (PUFAs), especially n-3 and n-6 fatty acids (FAs). These compounds have been shown to have both preventive and therapeutic effects against a range of diseases, including cancer, cardiovascular disorders, and inflammatory disorders [1]. While fresh fish is highly perishable, canned fish products offer the advantage of extended shelf life, making them a convenient alternative. Canned tuna, in particular, is one of the most widely recognizable fish products worldwide [2]. One common preservation method for canned fish uses oil as the liquid medium, which helps maintain product quality during storage by forming a barrier that limits oxygen exposure, rather than through bacteriostatic or bactericidal effects.

Among the various types of oil used in canned products, olive oil is often preferred and prominently featured on labels due to its association with a healthy diet [3]. Council Regulation (EEC) No 1536/92, which regulates the labeling and marketing of preserved tuna, specifies that if the product name includes the description “in olive oil”, only olive oil can be used in the product [4]. Given olive oil’s higher cost relative to cheaper vegetable oils (VOs) and fish meat’s ability to mask its distinctive flavor, it becomes a frequent target for adulteration. Consequently, the market standards for olive oil are more rigorous compared to other oils [5].

The quality of olive oil in the EU is defined according to analytical and sensory criteria established in Commission Regulations (EU) 2022/2104 [6] and 2022/2105 [7], while outside of the EU, the International Olive Council (IOC) standards apply [8]. According to EU Regulation 2022/2104, important indicators of olive oil authenticity are its FA composition, sterol composition, and levels of stigmastadienes [6]. Moreover, these quality and purity parameters must be analyzed following the official methods and standards of the IOC [8].

Olive oil is mainly composed of triglycerides with a high content of monounsaturated fatty acids (MUFAs), a relatively low content of PUFAs, and smaller amounts of other compounds, such as polyphenols (1–2%) of which 18–37% are biophenols, sterols (1–2%), and tocopherols (2–3%) [9]. Although the FA profile is a key factor in assessing olive oil quality and distinguishing it from other edible oils, the high MUFA content of oils like rapeseed oil complicates the detection of olive oil adulteration [10].

Studies have assessed the authenticity of oil in canned fish by analyzing FA composition, noting the challenge of confirming whether the oil is genuine due to lipid exchange between the fish and the oil, particularly the release of lipids such as *trans*-linoleic and *trans*-linolenic acids from the fish [3,11]. The sterol composition and the composition of triglycerides (TAG), expressed as the difference between the theoretical and actual content of TAG with 42 equivalent carbon numbers (ΔECN42), are also useful indicators of purity and for detecting adulteration of olive oils [12]. However, these official methods have some disadvantages, including the use of large volumes of chemicals and limited sensitivity in detecting low levels of adulteration, all of which highlight the need for more advanced techniques to detect olive oil adulteration.

Given the complexity of verifying the authenticity of high-quality foods such as olive oil, multiple analytical methods are required. One method for determining the authenticity of olive oils is the use of stable isotopes, as determined using Isotope Ratio Mass Spectroscopy (IRMS). The carbon isotopic ratio ^13^C/^12^C (expressed as δ^13^C) is a valuable tool for assessing the authenticity of vegetable products from plants with different photosynthetic pathways. It is related to the processes of atmospheric CO_2_ fixation, in which plant cells differentiate between the heavier ^13^C and the lighter ^12^C stable isotopes [13,14]. Since the olive tree follows the C3 photosynthetic pathway, it is possible to detect the potential addition of oils from plants from other photosynthetic pathways (C4 or CAM) [15]. This technique is now well-established for determining the authenticity of olive oil [16,17,18,19,20,21]. Moreover, one of the more recent advances in analytical chemistry has been the development of Compound-Specific Stable Isotope Analysis (CSIA) of organic compounds, which combines gas and/or liquid chromatographic separation with IRMS [22]. Although CSIA is just as technically demanding as official methods, it offers significantly improved specificity and sensitivity for detecting olive oil adulteration, making it a valuable complementary tool [23,24,25,26].

Combining lipid and isotopic composition analyses with chemometric methods offers several advantages for detecting adulteration in olive oil [20,27]. Studies have demonstrated that discriminant analysis, including partial least squares discriminant analysis (PLS-DA), enables the distinction between samples of olive oils based on their geographical origin [26,28], but it is rarely applied to reveal differences between authentic and adulterated samples. Moreover, among those studies that have investigated the authenticity of olive oil in canned fish products [3,5], none have applied CSIA.

This study assesses the authenticity of olive oil in canned tuna products from the Slovenian market by analyzing FA and sterol composition, ΔECN42 values, and the δ^13^C signatures of individual FAs. Given that detecting adulteration becomes increasingly complex at lower levels of adulterant oil [12], the study also compares the effectiveness of FA composition analysis and CSIA in identifying olive oil adulteration with 5–20% vegetable oil (VO) or hazelnut oil (HO), using a chemometric approach (PLS-DA). The findings aim to support analytical laboratories in more effectively assessing the quality and potential adulteration of olive oil in processed foods such as canned fish.

## 2. Materials and Methods

### 2.1. Samples

Ten cans of tuna packed in olive oil, as indicated on the label, were randomly selected from various manufacturers available in Slovenian supermarkets in 2007 (samples OL-01 to OL-10). Although the samples were collected and analyzed some time ago, the results remain relevant for evaluating the robustness and applicability of both official and advanced methods, especially in complex food matrices like canned tuna. The oil was carefully separated from the fish and immediately analyzed for the selected compounds. Three commercially available olive oil blends were also analyzed to determine the performance of both official and advanced methods in detecting olive oil adulteration. These included OO-A1, a blend of olive oil and various VOs; OO-A2, consisting of 70% virgin olive oil, 10% refined pomace olive oil, and 10% sunflower oil high in oleic acid; and OO-A3, comprising 80% virgin olive oil and 20% palm oil high in oleic acid.

Additionally, six canned tuna samples were selected to simulate the potential adulteration of olive oil. In each case, the olive oil was partially substituted with a VO blend or HO at 5%, 10%, and 20% mass fractions. Specifically, 5% VO was added to OL-08, 10% VO to OL-10, and 20% VO to OL-02, while 5% HO was added to OL-07, 10% HO to OL-03, and 20% HO to OL-04. Modified samples were analyzed immediately after substitution (Day 0) and again after 25 days of storage under dark and dry conditions at −18 °C, to prevent microbial contamination and ensure controlled slow lipid oxidation.

### 2.2. Chemicals and Reagents

All chemicals, potassium hydroxide, diethyl ether, propionitrile and methyl nonadecanoate were purchased from Sigma Aldrich (Taufkirchen, Germany). All solvents (analytical-reagent grade or GC grade), methanol, ethanol, heptane, hexane, were also from Sigma Aldrich (Taufkirchen, Germany). 

### 2.3. Official Methods for Determining Olive Oil Authenticity

#### 2.3.1. Fatty Acid Composition and Determination of the Difference Between Actual and Theoretical Content of Triacylglycerols (TAGs) with ECN 42 (ΔECN42)

The FA composition of olive oil was determined using in situ transesterification followed by GC-FID analysis (Agilent HP 6890 Series, Agilent Technologies, Santa Clara, CA, USA) in accordance with the method outlined in COI/T.20/Doc. No. 33/Rev 1 [29]. Briefly, fatty acid methyl esters (FAMEs) were prepared in heptane with a 2-M methanolic potassium hydroxide solution. Separation was achieved on a Supelco 2560 Capillary GC Column (100 m × 0.25 mm ID, df 0.20 μm; Supelco Inc., Bellefonte, PA, USA) with hydrogen as the carrier gas at a flow rate of 1 mL/min. The temperature program was as follows: 21 min at 180 °C; from 180 to 190 °C at 4 °C min^−1^; 11 min at 190 °C; from 190 to 220 °C at 10 °C min^−1^; 10 min at 220 °C; from 220 to 240 °C at 30 °C min^−1^; and 5 min at 240 °C. The FID temperature was 300 °C. FAs were assigned by comparing their retention times with those of a Supelco 37 Component FAME Mix reference standard (Supelco Inc., Bellefonte, PA, USA). The data are expressed as weight percentages of the total FAs. The measurement uncertainty was ±0.29% for the most abundant FA.

The official method for the ΔECN42 determination in olive oils is COI/T.20/Doc. No. 20/Rev. 4 [30]. The method calculates the absolute difference between the experimental values of TAGs with equivalent carbon number of 42 (ECN 42_HPLC_) obtained by HPLC and the theoretical value of TAGs with an equivalent carbon number of 42 (ECN42_theoretical_) calculated from the FA composition. The process can be divided into three phases: (1) determination of FA composition by capillary GC; (2) calculation of the theoretical TAG composition using ECN42; and (3) determination of ECN42 by HPLC.

Samples were extracted using an SPE silica gel cartridge preconditioned with hexane without allowing the level of the hexane to go below that of the sorbent. Approximately 0.12 g of the oil, dissolved in hexane, was loaded onto the cartridge and eluted with a hexane-diethyl ether solution under vacuum. The combined eluates were homogenized and divided into two volumes, one aliquot for GC and the second for TAG analysis by HPLC. The Agilent 1100 Series HPLC System (Agilent Technologies, Santa Clara, CA, USA) was equipped with a BinPump G1312A binary pump, a Agilent G1329A ALS thermostated autosampler (Agilent Technologies, Santa Clara, CA, USA), a thermostatted column compartment, and a refractive index detector operating at 35 °C. Separation was achieved using a Phenomenex LiCrospher/Superspher RP18 80A column (4 μm i.d., 250 × 4.0 mm). Propionitrile was used as the mobile phase.

#### 2.3.2. Determination of Sterol Composition

The sterol composition and content in the olive oil samples were determined by thin-layer chromatography and capillary gas chromatography with flame ionization detection (GC-FID) (Agilent HP 6890 Series, Agilent Technologies, Santa Clara, CA, USA) following the official method COI/T.20/Doc. No. 26/Rev.5 [31]. Briefly, sterols and triterpenic dialcohols were extracted from 5.0 g of oil in the presence of the internal standard α-cholestanol (0.2%, m/V). Sample preparation involved saponification with 2 M ethanolic potassium hydroxide, extraction of the unsaponifiable matter using diethyl ether, separation of sterols and triterpenic dialcohols from the unsaponifiable matter with thin-layer chromatography, derivatization to trimethylsilyl ethers, and analysis by gas chromatography. An Agilent HP 6890 Series (Agilent Technologies, Santa Clara, CA, USA), equipped with a Supelco SPB-5 Capillary GC Column (60 m × 0.53 mm ID, df 5.00 μm; Supelco Inc., Bellefonte, PA, USA) was used for FID detection. Sterols and triterpenic dialcohols were identified by comparing their relative retention times to those of the internal standard α-cholestanol.

### 2.4. Olive Oil Authenticity Using δ^13^C in Individual Fatty Acids

Compound-specific carbon isotope analysis of individual fatty acids was obtained using an Agilent 6890NGC-C system (Agilent Technologies, Santa Clara, CA, USA) coupled to an IsoPrime stable IRMS (VG Instruments Ltd., Manchester, UK) via a combustion interface under a continuous flow of helium. The interface consisted of a ceramic furnace with a copper oxide and platinum catalyst at 850 °C. The GC was operated with the same type of column and temperature program as the one used for the FAME analysis. The performance of the GC/C/IRMS system, including GC and combustion furnace, was evaluated after every 10 analyses by injecting a methyl nonadecanoate (C_19:0_) FAME with a δ^13^C value of −29.8‰. Precision, based on seven replicate analyses, ranged from ±0.1 to ±0.4‰. The isotopic shift due to the carbon introduced during methylation was corrected using the following mass balance equation:δ^13^C_FAME_ = [(C_n_ + 1) δ^13^C_FA_ − δ^13^C_MeOH_]/C_n_
where δ^13^C_FAME_, δ^13^C_FA_, and δ^13^C_MeOH_ are the carbon isotopic values of the FAME, FA, and methanol used for methylation, respectively, and C_n_ is the number of C atoms in the fatty acid [32].

### 2.5. Statistical Analysis

The XLSTAT software 2024.2.2 (Addinsoft, Long Island, NY, USA, 2024) was used for statistical analysis. Basic statistics included calculation of mean values, standard deviation, analysis of variance ANOVA, and a Kruskal–Wallis test. Differences were considered statistically significant at *p* < 0.05. Correlations between the variables were quantified using Pearson’s linear correlation coefficient (r).

Multivariate Partial Least Squares Discriminant Analysis (PLS-DA) was used to classify olive oil samples based on their FA and isotopic composition using the SIMCA (version 18.0.0.372, Sartorius Stedim Data Analytics, Umeå, Sweden). Candidates for discriminant markers were selected using loading plots, which illustrate the relationships between the formed groups and the variables, as well as by the Variable Importance in the Projection (VIP) values from the PLS-DA models, with a threshold set at VIP > 1.

## 3. Results and Discussion

### 3.1. Authenticity of Olive Oil in Canned Tuna from the Slovenian Market

#### 3.1.1. Fatty Acid Composition and ΔECN42

Appendix A presents a representative GC-FID chromatogram demonstrating the effective separation of fatty acid methyl esters in an olive oil sample from canned tuna. The content of the five major FAs from olive oils from different canned tuna is shown in Table 1. The results indicate that the FA composition of the analyzed samples corresponds to the typical composition of olive oil, as specified in EU Regulation 2022/2104 [6].

The sum of palmitic, palmitoleic, stearic, oleic, and linoleic acid accounts for 98% of the total FAs in all samples. All oils were highly unsaturated, with unsaturated FAs (USFAs) accounting for 81.0% to 84.1%. The oleic acid content varied from 65.3% in the OL-09 sample to 77.8% in the OL-02 sample. The linoleic acid content ranged between 5.3% and 14.0% across the samples. In contrast, the saturated fatty acids (SFAs) content varied between 14.2% and 17.4% in the analyzed oils. The ratio between USFAs and SFAs in analyzed oils ranged from 4.7 to 5.9, which is consistent with that reported by Al-Bachir et al. [33], who reported that the ratio between USFA and SFA was 4.9 for olive oil samples. The variation in FA composition of olive oil, especially in oleic, linoleic, and palmitic acids, could be attributed to various factors, for example, harvest time, climate conditions, origin, variety, farming practices, extraction methods, and storage conditions [34]. Furthermore, variations in FA contents may be due to the migration of FAs between the fish and the oil, as suggested by several studies [5,35,36].

Appendix A shows all identified FAs present in the olive oil samples in lower amounts. All FA levels complied with the limits set by EU Regulation 2022/2104, except for myristic acid, which exceeded the permitted value in five samples (OL-01, OL-03, OL-04, OL-07, and OL-09). Notably, myristic acid was only present in trace amounts (≤0.04% of total FAs).

The FA composition of three olive oil samples (OO-A1, OO-A2, OO-A3) blended with VO high in oleic acids was analyzed to evaluate whether the FA composition method is appropriate for assessing the authenticity of olive oil. Sample OO-A1 (Table 1) showed a significantly lower oleic and palmitic acid content and a higher linoleic acid content, suggesting it was blended with VOs rich in linoleic acid. The other two adulterated oils (OO-A2 and OO-A3) had FA contents within the limited values for authentic olive oil. The results suggest that the FA composition alone may not be sufficiently sensitive to detect adulteration with oils with a similar FA profile. Therefore, more rigorous monitoring and additional analytical methods are necessary to assess the quality and authenticity of olive oil and prevent fraud.

In addition, the ΔECN42 value was determined by first analyzing the FA composition, calculating the theoretical triglyceride composition based on ECN42, and determining the actual triglyceride composition using ECN42 measurements. Although the results (Table 1) show a slight variance between samples (0 to 0.15%), none of the samples exceeded the maximum permitted limit of 0.2% for ΔECN42, set by EU Regulation 2022/2104. These findings indicate no evidence of adulteration of olive oil in the canned tuna samples.

#### 3.1.2. Sterol Composition

Appendix A presents a representative GC-FID chromatogram demonstrating the effective separation of sterols in an olive oil sample from canned tuna. According to the results (Table 2), the most representative sterol in the samples is β-sitosterol (82.3–87.0% of total sterols), followed by Δ−5-avenasterol (2.5–7.9%) and campesterol (3.2–3.5%). β-Sitosterol and Δ5-avenasterol are considered the primary carriers of antioxidant activity and other beneficial health effects among the sterols in olive oil, due to their abundance and structural characteristics [37]. Stigmasterol, clerosterol, and Δ−5,24-stigmastadienol were found in similar contents, i.e., around 1% of the total sterols. Other identified sterols were present in lower amounts.

A higher campesterol-to-stigmasterol ratio has been reported as an indicator of olive oil quality [38]. In the present study, the campesterol/stigmasterol ratio ranged between 1.9 and 4.9, consistent with the literature [39], indicating the presence of good-quality olive oils. For apparent β-sitosterol, expressed as the sum of β-sitosterol and five other sterols formed by the degradation of β-sitosterol (sitostanol, Δ5,24-stigmastadienol, Δ−5,23-stigmastadienol, clerosterol, and Δ5-avenasterol), all of the samples contained more than the regulatory minimum level of 93%, indicating that the sum of the remaining sterols does not surpass 7%, thereby confirming the authenticity of the corresponding oils.

Furthermore, all olive oil samples from canned tuna contained total sterol levels above the minimum requirement of 1000 mg/kg specified in EU Regulation 2022/2104, with contents ranging from 1209 to 1687 mg/kg. The content of total sterols was also in agreement with the results of Miklavčič Višnjevec et al. [5], who also analyzed the olive oil from canned fish. We can conclude that all samples are authentic based on the determination of sterol content in the analyzed olive oil from canned tuna samples.

#### 3.1.3. Stable Carbon Isotope Composition of Fatty Acids

The δ^13^C values of five FAs from olive oils in canned tunas are presented in Table 1. The isotopic values of the four most abundant FAs ranged from −29.9‰ to −28.3‰, from −30.7‰ to −29.3‰, from −28.7‰ to −27.4‰, and from −29.9‰ to −28.2‰ for palmitic, stearic, oleic, and linoleic acids, respectively. The trend in the δ^13^C values aligns with the findings of Spangenberg & Ogrinc [24], who reported that initial isotopic fractionation occurs during the first elongation step, i.e., from palmitic to stearic acid, resulting in lower δ^13^C values in stearic acid. A second phase of isotopic fractionation occurs during the first desaturation step, resulting in increased δ^13^C values in oleic acid compared to stearic acid. The second desaturation step likely causes a slight decrease in δ^13^C values of linoleic acid, probably since this step occurs in chloroplasts rather than in the cytosol, where palmitic, stearic, and oleic acids are synthesized. The δ^13^C values of FAs also align with those reported by Spangenberg et al. [25], who found δ^13^C values of FAs in virgin olive oil ranged from −26.5 to −35.5‰, and with Bontempo et al. [28], who reported δ^13^C values for palmitic, stearic, oleic, and linoleic acids ranged between −34 and −26‰.

The δ^13^C values also show a positive correlation with each other. In particular, the δ^13^C of palmitic and stearic acids strongly correlated with the δ^13^C of oleic acid (r = 0.739 and r = 0.768, respectively) and linoleic acid (r = 0.789 and r = 0.684, respectively). The strongest correlation was found between the δ^13^C values of oleic and linoleic acid (r = 0.934). Spangenberg et al. [23] described a similar correlation between δ^13^C of palmitic acid (16:0) and oleic acid (18:1) for cold-pressed olive oil. They explained that a noticeable difference in these values suggests an admixture of cold-pressed virgin olive oils with refined olive oils or other VOs with different δ^13^C_16:0_ to δ^13^C_18:1_ ratios compared to authentic olive oil. The results also agree with those of Bontempo et al. [28], who reported a strong correlation between δ^13^C values of palmitic, stearic, oleic, and linoleic acids in authentic olive oils.

The results also show that the blended OO-A3 oil has significantly lower δ^13^C values of all FAs compared to olive oils from canned tuna. The OO-A2 sample showed a significantly lower δ^13^C value of oleic acid, while a higher δ^13^C value of linoleic acid was observed compared to olive oil from canned tuna. The ratio between δ^13^C_16:0_ and δ^13^C_18:1_ of OO-A1 and OO-A3 is similar to that of authentic olive oils. In contrast, the δ^13^C_16:0_ and δ^13^C_18:1_ values of OO-A2 differ significantly, with the δ^13^C_18:1_ value being 2.3‰ lower than the δ^13^C_16:0_ value, which indicates the blending of olive oil with other VOs.

The results from both official and advanced methods indicate that, for blended olive oil samples (OO-A1, OO-A2, OO-A3), a combination of analytical approaches is necessary to detect adulteration accurately.

### 3.2. Comparing Official and Advanced Methods for Detecting Olive Oil Authenticity in Canned Tuna

The study further evaluated the stable isotope method for detecting olive oil adulteration by comparing it with the commonly used FA composition method recommended in EU Regulation 2022/2104. Additionally, PLS-DA was applied to identify the most effective method for differentiating authentic and adulterated olive oil samples. For this purpose, six canned tuna samples containing genuine olive oil were adulterated with 5%, 10%, and 20% VO or HO. The samples were analyzed immediately (day 0) and after 25 days of storage.

#### 3.2.1. Fatty Acid Composition Analysis as an Official Method

Vegetable oil (VO) contained a significantly higher content of stearic and palmitoleic acid and a significantly lower content of oleic and linoleic acid (Table 3) compared to the limit values set by EU Regulation 2022/2104 for olive oil (Table 1). Consequently, when VO was added to olive oil from canned tuna, the content of oleic acid (C_18:1c_) decreased by 8.6% (5% VO), 17.4% (10% VO), and 37.1% (20% VO), compared to corresponding olive oil samples (OL-08, OL-10, OL-02, respectively) at day 0. The content of stearic acid (C_18:0_) increased, although not directly proportional to the percentage of added VO. Additionally, the MUFA/PUFA ratio decreased significantly with increasing VO content, reducing from 6.5 (5% VO) to 1.6 (20% VO) on day 0. However, it is important to note that different proportions of VO were added to olive oil from different canned tuna samples and that the FA composition varied among these oils.

The content of SFAs (C_16:0_ and C_18:0_) remained relatively stable during storage, regardless of the type and percentage of added oil (Figure 1), consistent with the known greater oxidative stability of SFAs compared to USFAs. Also, the USFA/SFA ratio remained constant in all samples with added VO (ranging from 5.3 to 5.8) over time. However, after 25 days of storage, samples blended with 10% and 20% VO had significantly higher oleic acid content and lower linoleic and linolenic acid content (Figure 1a–c). The degree of increase in oleic acid and decrease in linoleic and linolenic acids was proportional to the amount of VO added, with higher VO addition leading to greater initial PUFA levels and a more noticeable decrease during storage. The observed increase in the percentage of oleic acid is likely due to the stable levels of SFAs combined with a decrease in PUFAs, specifically linoleic and linolenic acids. Since linoleic and linolenic acids are highly reactive, they are prone to oxidation, especially in the presence of oxygen, light, and heat. Blending olive oil with VO, therefore, may reduce its stability, especially if the VO has a lower antioxidant content and contains less stable FAs [40]. These results agree with Morello et al. [40], who observed increased oleic acid content due to decreased PUFA content (linoleic and linolenic acids) after 12 months of storage of olive oil. The authors also suggested that the decrease in PUFA content during storage could be due to oxidative reactions at the double bonds, with reaction rates depending on the number of double bonds in the carbon chain [40].

Hazelnut oil (HO) has a similar FA composition to olive oil, with the main difference being a lower content of palmitic acid than the limit set by EU Regulation 2022/2104 for authentic olive oil (Table 3). Adding HO to olive oil samples resulted in a lower abundance of palmitic acid and a higher abundance of linoleic acid compared to the corresponding olive oil. Unlike the addition of 10% and 20% VO, adding HO did not result in significant changes in FA content after 25 days of storage, suggesting greater stability of the FAs (Figure 1d–f).

#### 3.2.2. Stable Carbon Isotope Analysis of Fatty Acids as an Advanced Method

When different oils share similar FA profiles, detecting olive oil adulteration based solely on FA composition becomes unreliable. To address this problem, the present study employed stable carbon isotope analysis of individual FAs to improve detection sensitivity. The effectiveness of CSIA depends on the magnitude of δ^13^C differences between the oils.

The findings from this study indicate that CSIA can detect even a 5% addition of VO or HO, primarily due to the lower δ^13^C values of FAs in these oils compared to those in the olive oil samples (Table 3). Blended olive oil samples exhibited significantly lower δ^13^C values for oleic acid compared to authentic olive oil samples from canned tuna (Table 1 and Table 3). In addition to the common practice of blending virgin olive oil with lower-quality oils (e.g., pomace olive oil) or more economical alternatives such as rapeseed oil, thermally induced degradation processes, e.g., deodorization and steam washing, have also been shown to alter both the isotopic signature and FA profile of the oil. Several other factors can influence the isotopic and FA composition of olive oils, including the production year, storage duration (due to the oxidation of unsaturated fatty acids), botanical variety, and the maturity of olives at harvest [16].

After 25 days of storage, δ^13^C_18:1_ and δ^13^C_18:2_ values of adulterated samples were higher compared to day 0, regardless of the percentage of VO or HO added (Table 3). This observation aligns with findings from Spangenberg & Ogrinc [24], who reported that olive oil samples stored for one year at ambient temperature showed an isotopic shift toward less negative δ^13^C values. This ^13^C enrichment was attributed to thermal and oxidative alteration during storage, likely due to the preferential release of isotopically lighter, thermally and chemically less stable lipid moieties. Olive oil is known to undergo slow oxidation over time, which could lead to carbon isotope fractionation, potentially resulting in an enrichment of ^13^C in oleic acid and a shift toward less negative δ^13^C values [23,24,25]. Moreover, the presence of VO or HO in adulterated samples may speed up the oxidative process due to differences in their antioxidant profiles and PUFA content [41].

#### 3.2.3. Partial Least Squares Discriminant Analysis (PLS-DA)

For PLS-DA, three models were built using 18 olive oil samples from canned tuna (6 authentic, 6 adulterated with VO, and 6 adulterated with HO), with six variables representing FAs and/or five variables representing δ^13^C values of individual FAs. Figure 2, Figure 3 and Figure 4 show the PLS-DA score plots for the three scenarios.

In the first scenario, only the content of FAs was considered (Figure 2a). The cross-validated correlation coefficient (Q2, −0.15) indicated a poor predictive ability for a two-component model with an accuracy of 0.89 and a correlation coefficient (R2) of 0.37. The two-dimensional score plot (Figure 2a) did not allow a clear separation between the authentic and blended olive oil groups, explaining 79.9% of the variance. The most relevant variables for the model (VIP > 1, variable importance in projection) were the levels of *cis*-oleic, *trans*-oleic, stearic, and linoleic acids (Figure 2b). Further evaluation of the model’s classification performance was conducted using the Receiver Operating Characteristic (ROC) curve, which yielded an Area Under the Curve (AUC) of 0.972 (Appendix A), indicating strong discriminatory power.

In the second scenario, δ^13^C values of FAs were used for discriminating between samples. The cross-validated correlation coefficient (Q2, 0.79) indicated the best predictive ability for a two-component model, with an accuracy of 1.00 and a correlation coefficient (R2) of 0.88. The score plot (Figure 3a) effectively separates the authentic and blended olive oil groups, accounting for 79.6% of the variance. The PLS-DA plot showed that the CSIA method provides significantly better separation than the FA composition alone. The most relevant variables for the model (VIP > 1) were δ^13^C_18:1_, δ^13^C_18:0,_ and δ^13^C_18:2_ (Figure 3b). The ROC curve analysis validated the model’s classification performance, with an AUC value of 1.0 (Appendix A), reflecting perfect discrimination between classes.

In the third scenario, FA composition and δ^13^C values of the FAs were used for sample discrimination. The cross-validated correlation coefficient (Q2, 0.72) indicated a high predictive ability for a two-component model, with an accuracy of 1.00 and a correlation coefficient (R2) of 0.85. The PLS-DA score plot (Figure 4a) effectively separated authentic olive oil from blended olive oil samples, accounting for 67.6% of the variance. The most relevant variables for the model (VIP > 1) were δ^13^C_18:1_, δ^13^C_18:0_, and δ^13^C_18:2_ (Figure 4b). The ROC curve analysis validated the model’s classification performance, with an AUC value of 1.0 (Appendix A), reflecting perfect discrimination between classes. The authentic olive oil samples are located on the left side of the plot, characterized by a higher content of oleic acid and higher δ^13^C values of FAs compared to the blended mixtures. The adulterated olive oil samples appear on the right side of the plot, distinguished by elevated linoleic acid levels and more depleted δ^13^C values of FAs compared to authentic samples.

Across all three PLS-DA models, δ^13^C values of the FAs are the key discriminative variables separating the sample groups. This finding suggests that CSIA is more effective for detecting adulteration compared to using FA composition alone. Although Scenario 3 includes more variables, the better performance of Scenario 2 is due to the higher discriminatory power of δ^13^C values of FAs compared to FA composition. However, the combined approach (Scenario 3) also demonstrated strong predictive ability and can be effectively used for reliable chemometric discrimination between authentic and adulterated oils.

## 4. Conclusions

This study highlights the potential of combining CSIA with FA profiling for authenticating olive oil. It demonstrates how integrating official and advanced methods with multivariate statistical approaches can enhance the detection of adulteration, providing more comprehensive insights than official methods alone. While FA composition analysis struggled to differentiate adulterants with similar FA profiles, using δ^13^C values of individual FAs significantly improved detection, although distinguishing between VO and HO blended samples remains challenging.

A comprehensive methodological approach, especially one that utilizes advanced methods, such as CSIA combined with chemometric techniques, offers a promising way to ensure faster and more accurate detection of food authenticity. Incorporating additional stable isotope ratios (δ^2^H and δ^18^O), as well as extending CSIA to other compounds such as sterols and polyphenols, could further refine detection accuracy and improve discrimination of adulterated samples. These advanced methods hold significant promise for enhancing the reliability and precision of food authenticity assessments across a wide range of food products.

## Figures and Tables

**Figure 1 foods-14-02667-f001:**
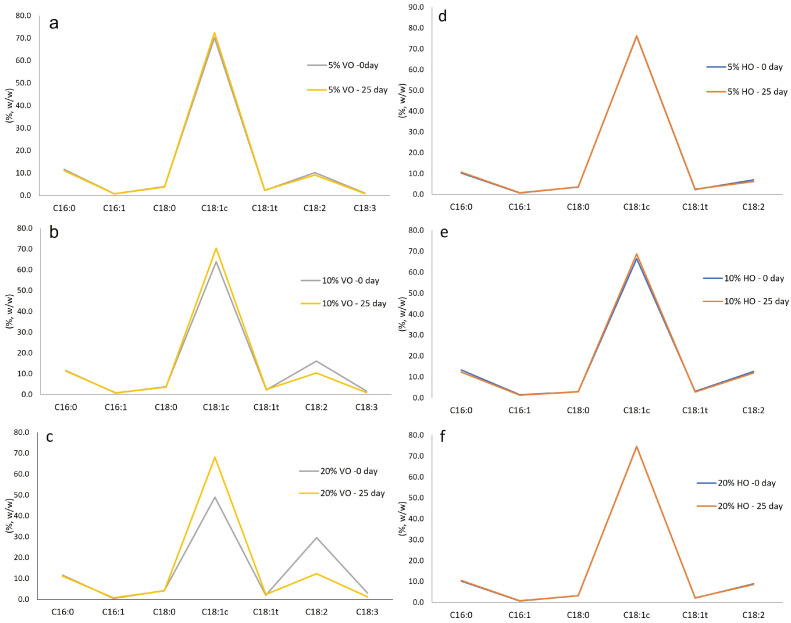
The fatty acid content in olive oil from canned tunas blended with vegetable oil (VO) (**a**–**c**) or hazelnut oil (HO) (**d**–**f**) at 5%, 10%, and 20%, respectively, analyzed on day 0 and after 25 days of storage. C16:0—palmitic acid; C16:1—palmitoleic acid; C18:0—stearic acid; C18:1c—*cis*-oleic acid; C18:1t—*trans*-oleic acid; C18:2—linoleic acid; C18:3—linolenic acid.

**Figure 2 foods-14-02667-f002:**
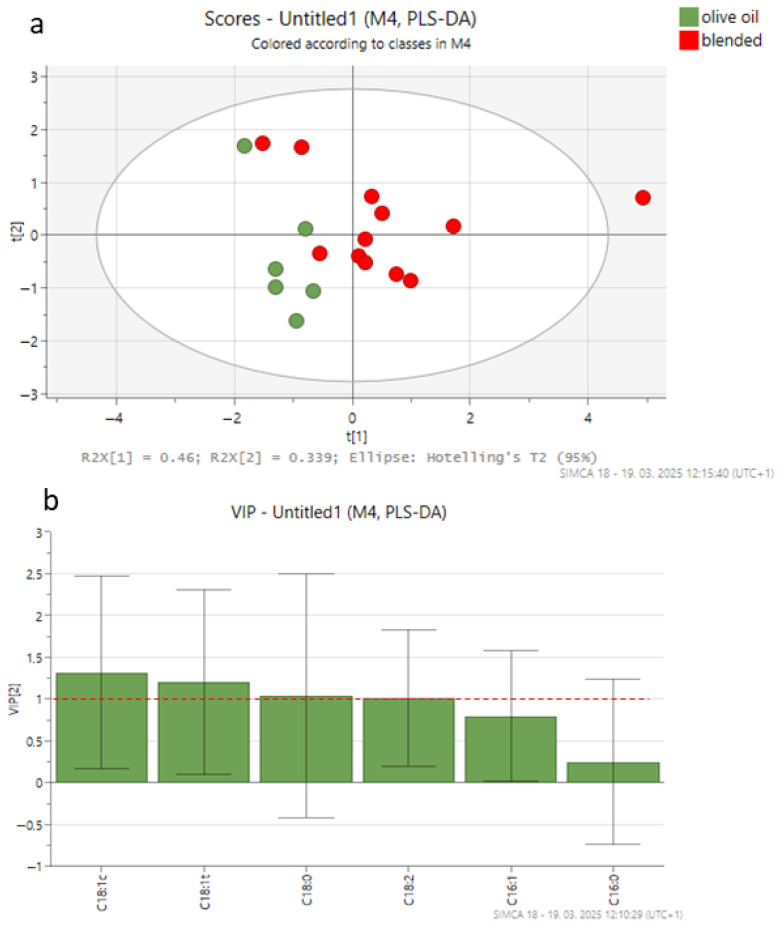
PLS-DA score plot (**a**) and VIP values (**b**) in the pairwise comparisons between different groups of olive oil from canned tuna (authentic olive oil; blended olive oils with vegetable oil (VO) or hazelnut oil (HO)) derived from fatty acid composition data. The ellipse on the score plot represents the 95% confidence interval.

**Figure 3 foods-14-02667-f003:**
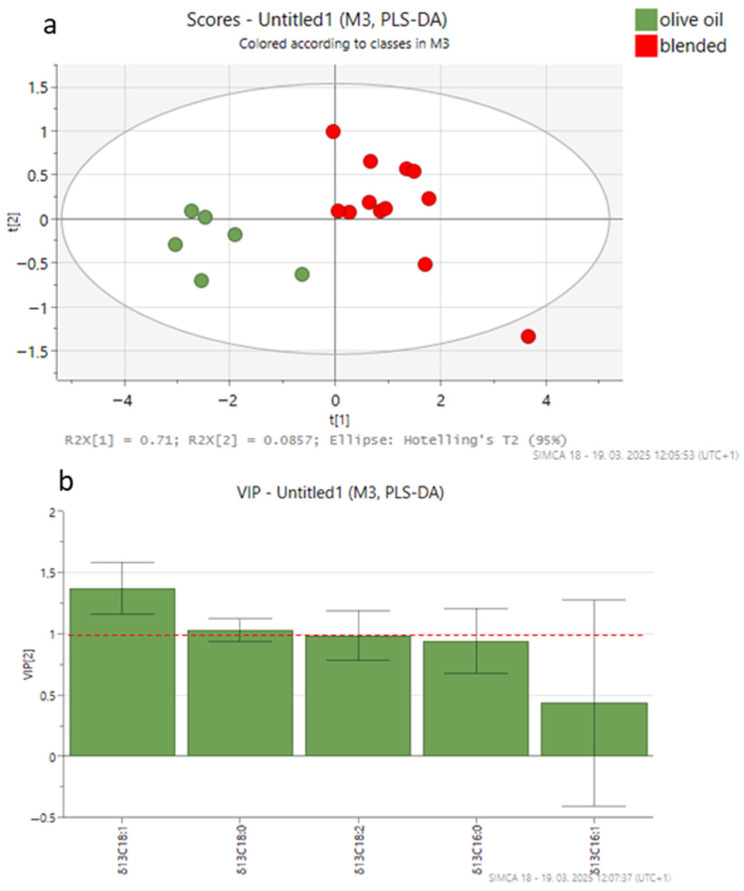
PLS-DA score plot (**a**) and VIP values (**b**) in the pairwise comparisons between different groups of olive oil from canned tuna (authentic olive oil; blended olive oils with vegetable oil (VO) or hazelnut oil (HO)) derived from carbon isotopic composition data. The ellipse on the score plot represents the 95% confidence interval.

**Figure 4 foods-14-02667-f004:**
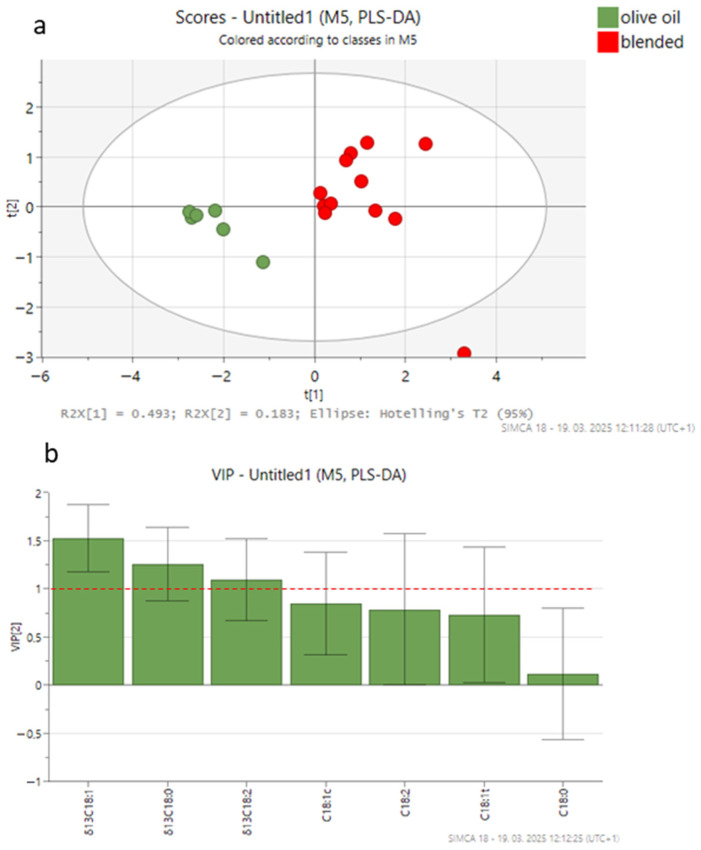
PLS-DA score plot (**a**) and VIP values (**b**) in the pairwise comparisons between different groups of olive oil from canned tuna (authentic olive oil; blended olive oils with vegetable oil (VO) or hazelnut oil (HO)) derived from fatty acid and carbon isotopic composition data. The ellipse on the score plot represents the 95% confidence interval.

**Table 1 foods-14-02667-t001:** Mean values for five dominant fatty acids, ΔECN42, and δ^13^C values of fatty acids in canned tuna olive oil (OL-01–OL-10) and in three blended olive oils (OO-A1–OO-A3).

		Fatty Acids (%, *w*/*w*)					δ^13^C (‰, VPDB)				
Sample	ΔECN42	Palmitic	Palmitoleic	Stearic	Oleic	Linoleic	Palmitic	Palmitoleic	Stearic	Oleic	Linoleic
**Limits ***	**≤0.2**	**7.00–20.0**	**0.3–3.5**	**0.5–5.0**	**55.0–85.0**	**2.5–21.0**					
OL-01	0.08	12.19	1.31	2.95	71.98	9.86	−29.4	−29.5	−30.1	−28.2	−29.6
OL-02	0.01	10.48	0.91	3.85	77.8	5.34	−29.4	−42.4	−30.0	−27.7	−29.0
OL-03	0.13	13.16	1.58	2.74	68.53	12.35	−29.9	−31.0	−30.6	−28.7	−29.9
OL-04	0.15	11.38	1.03	3.14	74.34	8.3	−28.9	−28.0	−29.8	−27.9	−29.0
OL-05	0.13	11.40	0.97	3.19	75.14	7.55	−28.8	−35.3	−29.9	−27.5	−28.4
OL-06	0.09	15.11	1.93	2.25	65.33	13.69	−29.1	−30.1	−30.3	−27.9	−29.1
OL-07	0.01	11.02	0.96	3.47	77.2	5.73	−28.4	−28.6	−29.7	−27.6	−28.6
OL-08	0.01	10.72	0.86	3.69	76.74	6.29	−29.2	−38.3	−29.3	−27.5	−28.3
OL-09	0.15	14.86	1.72	2.36	65.43	13.96	−29.1	−29.7	−30.7	−28.3	−29.1
OL-10	0	10.78	0.94	3.39	77.34	5.88	−28.3	−30.2	−30.0	−27.4	−28.2
OO-A1		6.51		3.58	31.29	57.64	−29.1		−29.8	−28.5	−28.0
OO-A2		10.00		3.91	75.47	7.75	−28.7		−29.9	−31.0	−27.7
OO-A3		17.08		3.62	66.61	9.92	−31.5		−32.9	−31.5	−30.8

* Limits established by the current EU Regulation 2022/2104 for olive oil.

**Table 2 foods-14-02667-t002:** Mean values for all identified sterols of olive oil from canned tuna samples (OL-01–OL-10).

	Sterols (%, *w*/*w*)																
Samples	Cholesterol	Brassicasterol	24-methylene-cholesterol	Campesterol	Campestanol	Stigmasterol	Δ−7-campesterol	Δ−5,23- stigmastadienol	Clerosterol	β-sitosterol	Sitostanol	Δ−5-avenasterol	Δ−5,24- stigmastadienol	Δ−7-stigmastenol	Δ−7-avenasterol	App. β-sitosterol	Total sterols (mg/kg)
**Limits ***	**≤0.5%**	**≤0.1%**		**≤4.0%**		**≤campesterol**	**<0.5%**			**<93.0%**				**≤0.5%**		**≥93%**	**≥1000 mg/kg**
OL-01	0	0	0.39	3.53	0.08	0.98	0	0.27	1.16	84.44	0.45	6.99	0.86	0.39	0.44	94.17	1650
OL-02	0	0	0.22	3.4	0.09	0.69	0	0.23	1.04	86.44	0.39	6.21	0.7	0.26	0.34	95.01	1417
OL-03	0	0.1	0.65	3.45	0.08	0.84	0	0.53	1.04	85.52	0.45	5.29	1.04	0.46	0.56	93.87	1681
OL-04	0	0	0.30	3.38	0.10	1.47	0	1.19	1.38	85.39	0.72	2.83	2.19	0.49	0.54	93.7	1240
OL-05	0	0	0.19	3.46	0.09	1.46	0	1.60	1.45	85.56	0.62	2.49	2.34	0.47	0.28	94.06	1261
OL-06	0	0	0.65	3.52	0.12	1.83	0	0.18	0.97	82.27	0.92	7.68	0.99	0.41	0.73	93.01	1687
OL-07	0	0	0.39	3.22	0.17	0.82	0	0.35	0.99	85.34	0.48	5.89	0.96	0.37	1.02	94.01	1484
OL-08	0	0	0.18	3.21	0.12	0.75	0	0.59	1.47	85.70	0.68	5.34	0.76	0.30	0.89	94.54	1380
OL-09	0	0	0.27	3.53	0.14	1.46	0	0.26	0.95	83.09	0.60	7.85	0.66	0.48	0.71	93.41	1667
OL-10	0	0	0.13	3.23	0.09	0.89	0.15	0.65	0.51	86.99	0.43	5.22	0.62	0.36	0.73	94.42	1209

* Limits established by the current EU Regulation 2022/2104 for olive oil.

**Table 3 foods-14-02667-t003:** Mean values for all identified fatty acids and carbon stable isotope (δ^13^C) of fatty acids in olive oil (OO) from canned tuna samples blended with vegetable oil (VO) mixture and hazelnut oil (HO) in varying proportions (5–20%), analyzed at day 0 and after 25 days of storage.

		Fatty Acids (%, *w*/*w*)							δ^13^C (‰, VPDB)					
Sample		Palmitic	Palmitoleic	Stearic	Oleic	*t*-Oleic	Linoleic	Linolenic	Palmitic	Palmitoleic	Stearic	Oleic	Linoleic	Linolenic
VO		12.0	4.2	22.5	52.5	1.5	0.6	3.1	−31.0	−32.4	−30.9	−30.8	−33.4	−33.2
HO		6.5		3.1	79.7	1.4	9.3		−31.7		−32.1	−30.6	−30.3	
**Day 0**														
OL-08	5% VO + 95% OO	11.6	0.7	4.0	70.2	2.3	10.2	1.0	−29.7	−31.0	−31.4	−31.5	−30.1	−31.7
OL-10	10% VO + 90% OO	11.6	0.7	3.8	63.9	2.3	16.1	1.6	−29.6	−30.6	−31.8	−30.9	−30.7	−31.8
OL-02	20% VO + 80% OO	11.7	0.5	4.2	49.0	2.0	29.6	3.1	−30.1	−30.9	−31.9	−32.1	−31.3	−31.7
OL-07	5% HO + 95% OO	10.3	0.7	3.6	76.0	2.3	7.1		−30.6	−30.1	−31.9	−31.8	−29.9	
OL-03	10% HO + 90% OO	13.3	1.5	2.9	66.6	3.1	12.7		−32.2	−30.3	−34.1	−31.2	−31.9	
OL-04	20% HO + 80% OO	10.2	0.7	3.3	74.7	2.2	9.0		−30.7	−30.3	−31.8	−32.0	−30.1	
**Day 25**														
OL-08	5% VO + 95% OO	11.0	0.8	3.8	72.4	2.4	9.1	0.9	−30.2	−30.2	−31.1	−30.3	−29.8	−30.8
OL-10	10% VO + 90% OO	11.3	0.8	3.6	70.4	2.5	10.4	1.0	−29.7	−29.7	−31.2	−31.0	−28.9	−31.4
OL-02	20% VO + 80% OO	11.1	0.7	4.1	68.2	2.4	12.3	1.3	−30.3	−30.4	−31.1	−30.9	−30.1	−31.4
OL-07	5% HO + 95% OO	10.7	0.8	3.5	76.3	2.5	6.2		−29.8	−31.9	−31.5	−30.1	−29.7	
OL-03	10% HO + 90% OO	12.3	1.3	2.9	68.8	2.8	11.9		−31.1	−31.6	−32.4	−30.8	−30.7	
OL-04	20% HO + 80% OO	10.6	0.8	3.2	74.5	2.2	8.7		−30.6	−31.0	−31.8	−30.9	−29.9	

## Data Availability

The original contributions presented in the study are included in the article/Appendix A, further inquiries can be directed to the corresponding author.

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
