# Peer review of "Chemometric Evaluation of Official and Advanced Methods for Detecting Olive Oil Authenticity in Canned Tuna"

_foods, 2025, doi:10.3390/foods14152667_

Round 1
Reviewer 1 Report
Comments and Suggestions for Authors
This study presents an evaluation of olive oil authenticity in canned tuna by integrating several methods (FA profiling, AECN42, sterols) and advanced techniques (CSIA-IRMS of δ13C), combined with chemometrics (PLS-DA).
Comments:
1) No original test spectra are provided, only tables and figures after calculating.
2) In line 257, the authors demonstrated as “A higher campesterol/stigmasterol ratio has been reported as an indicator of olive oil quality [38]. In this study, the campesterol/stigmasterol ratio ranged between 1.9 and 4.9 in samples, consistent with other studies [42]”
Other studies seem not support this conclusion because there is NO reference [42].
3) In line 380, the authors wrote “Unlike VO, adding HO did not result in significant changes in FA content after 25 days of storage, suggesting greater stability of the FAs.”
However, in Figure 1, VO did not result in significant change if adding 5%. Also, no HO data was provided.
And, why the data in Figure 1 only changes in the term of c18:1c and c18:2? And why all three orange lines looks so similar considering different percent of VO is added?
4) In line 411,”Olive oil is known to undergo slow oxidation over time, leading to carbon isotope fractionation, which results in an enrichment of ¹³C in oleic acid and a shift toward less negative δ¹³C value.”
Experimental data is necessary. carbon isotope fractionation may not occur in simple oxidation in room temperature.
5) PLS-DA data has been provided by benchmarking differnt factors, from (a) content of FAs, then (b) δ13C values of FAs, finally (c) FA composition and δ13C values of the FAs. It is quite strange why b is better than c, while in general case, more factors behave better than only one if it follows linear fitting.
Reviewer 2 Report
Comments and Suggestions for Authors
The authors did a good work from an experimental point of view, and I recommend the article for publication after some major revisions.
More specific:
L110: The samples were collected in 2007, yet the manuscript is dated 2025. Please clarify the rationale for analyzing samples 18 years post-collection. Were these samples preserved continuously under laboratory conditions, and how representative are they of typical market products or consumer environments?
L122: The manuscript mentions storage under dry and dark conditions at –18 °C. Could the authors justify how this reflects real-world consumer handling? If not, this may limit extrapolation of oxidation and isotope shift data to retail scenarios.
L208: Have the authors considered extending CSIA to polyphenols or sterols? These could offer compound-specific isotope markers with greater specificity, potentially enhancing differentiation of adulterants.
L376: The authors acknowledge difficulty in distinguishing hazelnut oil from vegetable oil adulteration. Consider discussing additional isotopes such as δ²H and δ¹⁸O or sterol profiling to improve resolution. Are there plans to integrate these in future studies?
L395: Did the study explore untargeted metabolomics or spectroscopic techniques (e.g., NIR, FTIR) as complementary tools? These might strengthen discriminative power, especially for adulterants with similar FA profiles.
L412: The observed shift in δ¹³C values is attributed to oxidation. A concise mechanistic explanation, perhaps referencing kinetic isotope effects, would reinforce this interpretation. Are there relevant studies the authors could cite?
L417: While CSIA successfully identifies adulteration down to 5%, the manuscript would benefit from numerical sensitivity values or ROC/AUC metrics from the PLS-DA models to substantiate predictive power.
L466: Correct the ''Figure 5'' by ''Figure 4''.
L511: References [6] & [7]: These are cited in the text but lack consistent formatting in the reference section. Please ensure full bibliographic details are included according to journal standards.
Author Response
See attachement.

Round 2
Reviewer 1 Report
Comments and Suggestions for Authors
All my concerns have been well addressed
Reviewer 2 Report
Comments and Suggestions for Authors
The paper has been revised according to the suggestions and criticisms of the reviewers. In this revised version, the paper has improved its quality and I recommend the article for publication.